# High-density, highly sensitive sensor array of spiky carbon nanospheres for strain field mapping

Shuxing Mei[1,4], Haokun Yi[1,4], Jun Zhao[1], Yanting Xu[1], Lan Shi[1], Yajie Qin [2], Yizhou Jiang [2], Jiajie Guo [3], Zhuo Li [1] ✉ & Limin Wu [1] ✉

While accurate mapping of strain distribution is crucial for assessing stress concentration and estimating fatigue life in engineering applications, conventional strain sensor arrays face a great challenge in balancing sensitivity and sensing density for effective strain mapping. In this study, we present a Fowler-Nordheim tunneling effect of monodispersed spiky carbon nanosphere array on polydimethylsiloxane as strain sensor arrays to achieve a sensitivity up to 70,000, a sensing density of 100 pixel cm$^{-2}$, and logarithmic linearity over 99% within a wide strain range of 0% to 60%. The highly ordered assembly of spiky carbon nanospheres in each unit also ensures high inter-unit consistency (standard deviation ≤3.82%). Furthermore, this sensor array can conformally cover diverse surfaces, enabling accurate acquisition of strain distributions. The sensing array offers a convenient approach for mapping strain fields in various applications such as flexible electronics, soft robotics, biomechanics, and structure health monitoring.

Strain field mapping provides a comprehensive distribution of strain across an entire structure, thus opens up numerous opportunities that cannot be achieved by conventional discrete strain sensors. It validates the outcomes of finite element analysis (FEA) simulations and enhances confidence in results[1,2], It enables timely detection of structural anomalies in structural health monitoring[3,4], and aids in health condition assessment by identifying heterogeneous deformations in human body[5]. Currently, optical methods are predominantly used for full-field strain mapping due to their high spatial resolution[6]. However, these methods are susceptible to lighting conditions and often require complex offline calculations to convert optical data into strain fields[7]. In this regard, stretchable strain sensor arrays have emerged as a promising alternative with higher reliability and lower computational complexity. But their current spatial resolution is insufficient for many practical applications[6]. For instance, although these sensor arrays can detect the presence of a crack, they are unable to describe the strain

gradient near it[8,9], as achieved by optical methods. Therefore, it is imperative to increase the sensing density to attain a desirable spatial resolution for strain field mapping.

Nonetheless, the design of strain sensing arrays faces a great challenge in balancing sensing density and sensitivity, which are both essential parameters for such arrays. Current flexible strain sensors primarily rely on three typical mechanisms to obtain high sensitivity: crack propagation[10–12], contact resistance[13–15] and percolation mechanism[16,17]. Crack propagation sensors can only monitor deformation in the direction perpendicular to the crack. To enable omnidirectional sensing, multiple sensors are required, often arranged in a rosette-shaped configuration[18,19] or multiple-stack structures first[20,21]. However, this approach inevitably increases the size of the sensing unit and decreases the sensing density. Sensors based on contact resistance and percolation mechanisms depend on the breakage of conductive pathways for sensing. Since the distribution of conductive pathways and their alteration

[1]Department of Materials Science and State Key Laboratory of Molecular Engineering of Polymers, Fudan University, 220 Handan Rd., Shanghai 200433, China. [2]Micro-Nano System Center, School of Information Science and Technology, Fudan University, 220 Handan Rd., Shanghai 200433, China. [3]State Key Laboratory of Intelligent Manufacturing Equipment and Technology, School of Mechanical Science and Engineering, Huazhong University of Science and Technology, Wuhan, Hubei, China. [4]These authors contributed equally: Shuxing Mei, Haokun Yi. ✉e-mail: zhuo_li@fudan.edu.cn; lmw@fudan.edu.cn

during stretching vary randomly among different sensing units, the sensor size must be sufficiently large to minimize the response variation and ensure inter-unit consistency. This also restricts the sensing density. As a result, the reported density of flexible strain sensing array remains relatively low (Supplementary Table 1)[14,22–36].

To address this issue, we present a sensing mechanism based on Fowler-Nordheim (F-N) tunneling effect in a high-density sensing array composed of orderly packed, monodispersed spiky carbon nanospheres (SCNs) (Fig. 1a). We have observed F-N tunneling effect in a transparent pressure sensor consisting of SCNs/ polydimethylsiloxane (PDMS) with low concentration (<1.5 wt.%), which led to orders of magnitudes change in electrical current with minimal alternations in inter-particle distance[37]. Herein, we further found that the SCNs were able to assemble and be patterned into strain sensor arrays that not only offer high sensitivity in all directions, but also effectively mitigate randomness among sensing units to ensure inter-unit consistency even at small unit sizes. Accordingly, the fabricated strain sensor arrays on PDMS can achieve a sensing density of 100 pixel cm$^{-2}$ with standard deviation (SD) no more than 3.82% and a gauge factor (GF) up to 70,000. Additionally, these arrays exhibit 99% logarithmic linearity from 0% to 60% strain for each sensing unit, which can also offer conformal coverage on diverse surfaces, and robustness against various forms of deformations (Fig. 1b). With these superior properties, the sensor array film can be easily attached to different objects, providing detailed strain distribution akin to FEA simulations (Fig. 1c–e). The successful demonstration of strain field mapping in various scenarios, including stretching, crack propagation, and indentation, highlights the immerse potential of this sensing film for a wide range of applications.

## Results

### Fabrication of the sensing film

The fabrication process of the sensor array film is illustrated in Fig. 2a. SCNs were synthesized through seeded swelling oxidation polymerization method. The monodispersed polystyrene (PS) seeds with a diameter of 500 nm were swollen by aniline (ANI) monomers, and the polymerization was initiated by $Fe(NO_3)_3$ to obtain PS@ polyaniline (PANI) particles where generated PANI forms conductive spikes[38]. Subsequent carbonization process at 1200 °C removed the PS cores and converted PANI spikes into carbon spikes. The detailed structures of SCNs can be clearly seen in Fig. 2b. The synthesized SCNs exhibit uniform morphology with a diameter of 500 nm and a spike length of 105 nm.

The obtained SCNs were then dispersed in ethanol and sprayed on water. Due to their low density and high hydrophobicity, SCNs could float on water. The dissolution of ethanol in water induced a decrease in the local surface tension, creating a surface tension gradient near the droplet. This gradient induced Marangoni flow that pushed the spheres to orderly assemble into a continuous film (Supplementary Movie 1). A porous sponge was then put inside the SCNs covered water, and the floating spheres immediately contracted toward the opposite side due to the capillary force, which finally led to a compact structure[39,40]. The contrasting surface energies of SCNs and water facilitated the film transfer from water surface to various substrates, regardless of their composition (Supplementary Fig. 1) or surface topography (Supplementary Fig. 2). Here we initially transferred it to a sacrificial polyvinyl alcohol (PVA) substrate, which was then patterned into a dense sensor array with fine pitch size in one step through laser-scribing. Subsequently, PDMS was spin coated on the patterned film. After curing of the

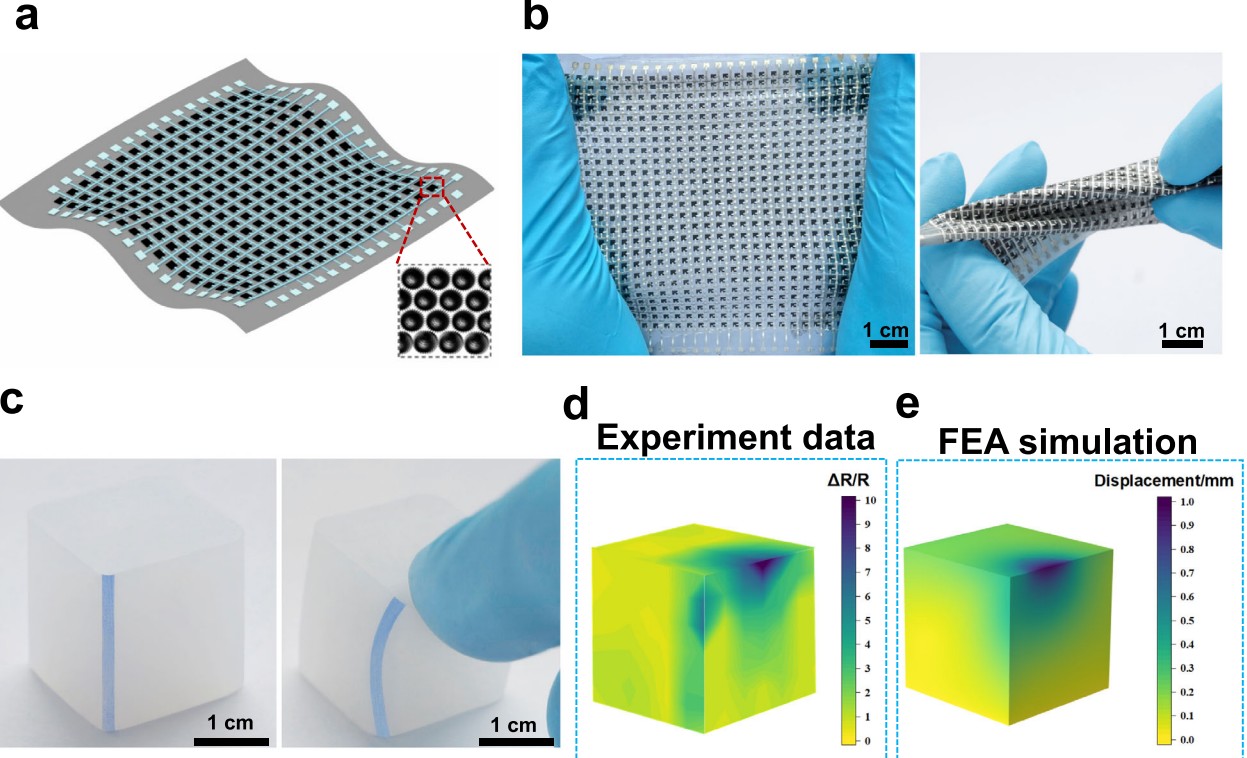

**Fig. 1 | Soft sensing films for strain field mapping. a** The sensing film is composed of ordered SCNs in the PDMS matrix. **b** The low modulus, small thickness and stretchability of the film can undergo various deformations, such as stretching (left), and twisting (right). **c** An elastomer cube before and after the pressing by a finger (the photos were taken without the sensor array to show a clear picture of the deformation). **d** The application of the sensing film on the cube can detect strain distribution on different surfaces of object. **e** The result is consistent with the FEA (finite element analysis) simulation. The results are shown in the initial state of the elastomer cube. In order to obtain a distinct representation of the sensing units, a 30 × 30 sensor array with a sensing density of 16 pixel cm$^{-2}$ was employed. Source data are provided as a Source Data file.

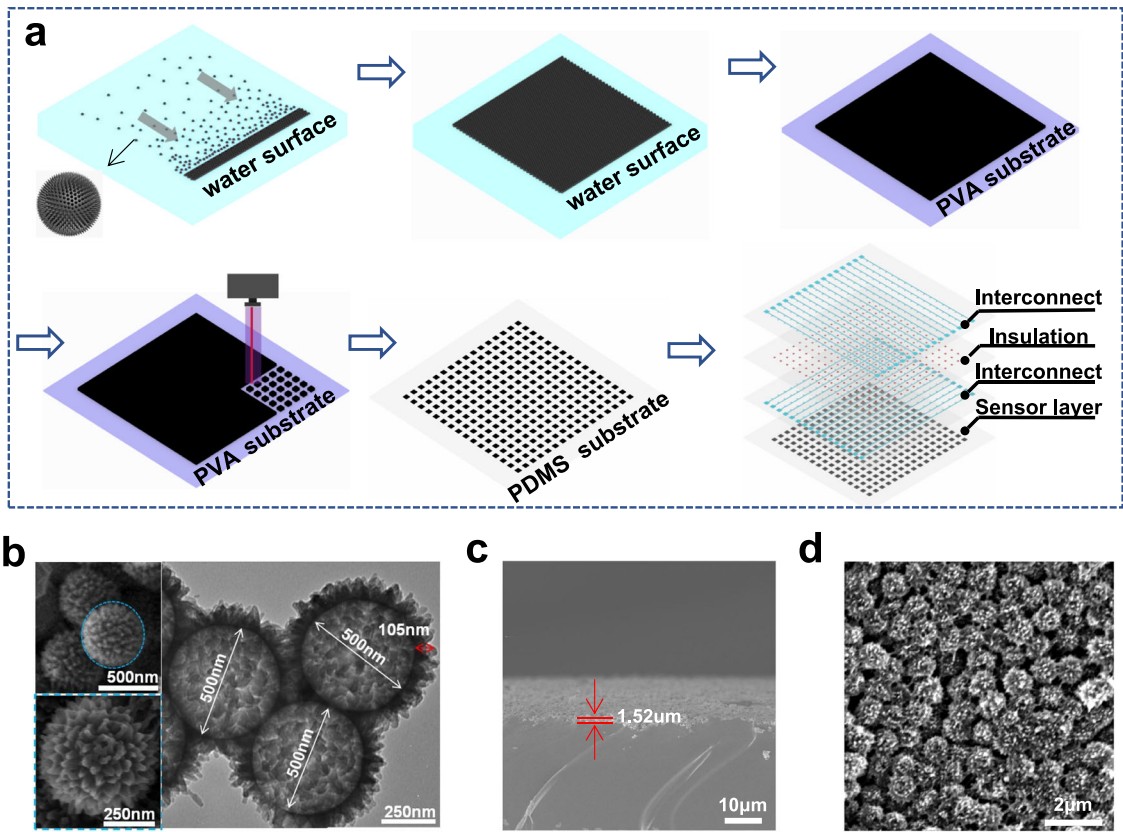

**Fig. 2 | Schematics of fabrication process and characterizations of the sensor array. a** Fabrication process of sensor array (PVA: polyvinyl alcohol; PDMS: poly-dimethylsiloxane). **b** SEM and TEM images of the SCNs. SEM images of the SCNs layer with (**c**) side and (**d**) top view.

PDMS, PVA was removed by dissolution in 95 °C water. Compared to the direct transfer to PDMS substrate, the additional step to PVA layer partially embedded the SCNs within PDMS, providing protection against scratching or detachment from the PDMS substrate (Supplementary Fig. 3). Scanning electron microscope (SEM) image confirms that the closely-packed SCNs layer can be maintained after the transferring and patterning process (Fig. 2c, d).

Finally, PDMS/silver (Ag) flakes composites were printed on the array layer as interconnect and pure PDMS as the insulation to prevent short circuits between the longitude and latitude conductive lines (Supplementary Movie 2). Because the interconnects and insulators had the same polymeric composition with the substrate, the inter-diffusion and entanglement of PDMS chains during curing merged all the components into a monolithic sensing film, which ensured homogeneous deformation during sensing. The entire array maintained a small thickness below 77 μm and a low modulus of 2.5 MPa (Supplementary Fig. 4), which can form conformal coverage on the target surface during strain measurement.

## Performance of a sensor unit

Figure 3a characterizes a typical resistive response of a sensing unit to the uni-axial strain $\varepsilon$. Resistance exhibits an exponential increase with the applied strain, yielding a GF as high as 70,000 at 60% strain. The GF is defined as GF = $\Delta R / \varepsilon R_O$, where $\Delta R$ represents the resistance change of sensor during deformation, and $R_O$ represents the initial resistance of the sensor. Once the strain exceeds 60%, the resistance of sensor is so high that it reaches the upper limit of our high resistance meter (Supplementary Fig. 5), thus, we only include data up to 60% strain. In comparison, traditional metal-based strain gauges show a GF of only 2–5[41], while GF of soft strain sensors based on percolation mechanism and contact resistance change is usually a few hundreds to a few

thousands, respectively[14–16,42–48]. Sensors based on crack propagation mechanism show a compromise in sensitivity and stretchability. It can reach a GF value up to 85500 with a low stretchability of only 5%. When stretchability is over 100%, the GF value drops to only several hundred[11,12,49–52]. In contrast, our sensor exhibits both high sensitivity and large stretchability as compared to the previous reports (Fig. 3b and Supplementary Table 2). Besides, our sensor exhibits a response time of 80 ms (forward) and 110 ms (reverse) (Supplementary Fig. 6), which is comparable to other flexible strain sensors[53,54]. Accordingly, the sensor can be used to detect both large movements and subtle vibrations. As shown in Supplementary Fig. 7a, a single sensor unit was attached to a volunteer's neck. When the volunteer was nodding, raising, or shaking his head, the sensor would generate a specific pattern of signals corresponding to each action and the signals were repeatable when the same action was performed. Besides the large movements, the sensor could also distinguish subtle strains from neck muscles during speaking (Supplementary Fig. 7b). Specific patterns appeared when the volunteer pronounced different words, such as Man, Apple and Banana.

Besides its high sensitivity, another feature of the present sensor is the good log$R$-$\varepsilon$ linearity ($R^2 = 0.992$). Different from previous studies that the linear response was restricted in a certain strain region[49,55–57,47,58], the logarithmic linearity here extends through the entire strain range of 0–60%. It simplifies the calibration and signal processing, which becomes particularly important in a high-density sensor array, where the data processing can be time-consuming and impact the real-time reconstruction of strain field.

The high sensitivity and logarithmic linearity observed are believed to be attributed to the F-N tunneling effect of SCNs. In order to verify the existence of F-N tunneling effect, the resistance changes under applied voltage, increasing from 0.1 V to 200 V in increment of 0.1 V were recorded (Fig. 3c). Unlike ohmic conductors in which

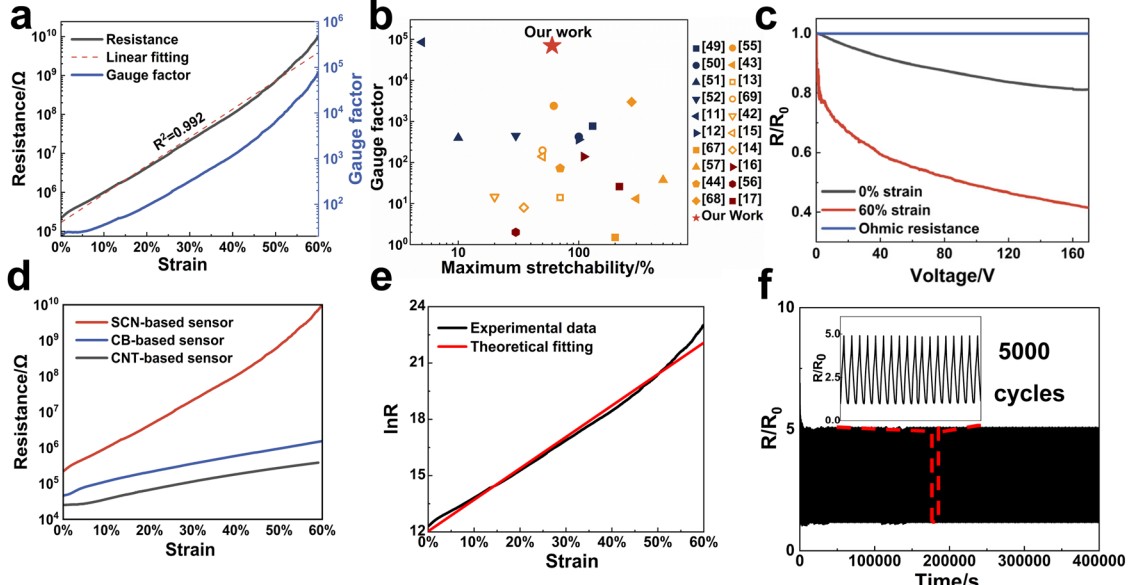

**Fig. 3 | Performances of the sensor unit based on SCNs/PDMS composite.**
**a** Resistance and gauge factor of the strain sensor versus the applied strain. **b** The gauge factor and maximum stretchability of SCNs based sensors (red star) as compared with previously reported crack (blue), contact (yellow), percolation (wine red) sensors[11–17,42,49–52,55–57,67–69]. **c** $R/R_O$ (Resistance change) curves of SCNs/PDMS sensor under different voltages and strains as compared with ohmic response. **d** Resistance as a function of the applied strain for sensors fabricated with CB (carbon black), CNT (carbon nanotube), and SCNs (spiky carbon nanospheres) as conductive materials. **e** The logarithm of the resistance value (ln$R$) experimentally obtained from the sensors fabricated with SCNs and the theoretical fitting curves based on the proposed model. **f** $R/R_O$ (Resistance change) of SCNs/PDMS sensor unit during 5000 stretch and release cycles under 10% strain, inset shows enlarged view of the recorded signals. Source data are provided as a Source Data file.

resistance remains unchanged with increasing voltage, our sensor exhibits a decrease in resistance as voltage increases. This is because the increased voltage reduces the tunneling barrier, which promotes the electron tunneling effect and leads to a larger tunneling current (i.e., a lower resistance). In addition, the decrease in resistance with voltage becomes even more pronounced when the applied strain reaches 60%. This observation may be attributed to a higher energy barrier for tunneling after further separation of neighboring SCNs at larger strain. The increased strain makes the effect of reducing the energy barrier through elevated applied voltage more significant. These results indicate the existence of F-N tunneling effect during the whole sensing range in SCNs/PDMS sensor. We also compared performances of the present sensor with those fabricated with other carbon materials under the same process (Supplementary Fig. 8). Carbon nanotube (CNT)- and carbon black (CB)-based sensors exhibit GF values of only 20 and 50 at 60% strain, respectively, consistent with previous reports[40] (Fig. 3d). Yet, GF of both sensors are three orders of magnitude lower than that of our SCNs/PDMS, highlighting the significance of the F-N tunneling effect arising from the spiky morphology of SCNs.

To gain deeper insights into the sensing behavior of our sensor, a model based on F-N effect model is proposed. The classical F-N model describes the relationship between current density $J$ and electric field strength $E_d$ of an insulating layer sandwiched between two electrodes[59]:

$$J = AE_d^2 \exp\left(\frac{B}{E_d}\right) \quad (1)$$

where $A$ and $B$ are both constants, $A$ is positive and $B$ is negative. Since our goal is to establish the relationship between current density $J$ and strain $\varepsilon$, the relationship between strain and electric field strength is needed. By taking consideration of electric field concentration effect induced by spike structure and geometrical relationships between two adjacent spheres, the electric field strength $E$ of a sphere spike applied

on the surface of the other sphere can be expressed as:

$$E = \frac{U}{(N-1)}\lambda(\varepsilon L_0 + d_0)^{-1} \quad (2)$$

where $U$ is the external voltage, $N$ is the number of SCNs in a conductive pathway, $\lambda$ is the concentration coefficient of the electric fields (detailed derivation can be found in Supplementary Notes 1), $L_O$ is the characterized length of sensor derived by geometrical relationships between two adjacent spheres (Supplementary Notes 2), $d_O$ is the distance of two spheres before stretching. Thus, for a conductive pathway containing $N_i$ SCNs, the current density can be expressed as:

$$J = A\left(\frac{U\lambda}{N_i-1}\right)^2 (\varepsilon L_0 + d_0)^{-2} \exp\left[\frac{B(N_i-1)}{U\lambda}(\varepsilon L_0 + d_0)\right] \quad (3)$$

The total current is supposed to be the addition of the current of all conductive pathways. However, the current is mainly determined by the several shortest pathways. Thus, $\frac{1}{N_i-1} \approx \frac{1}{N_O-1}$, where $N_O$ is the number of SCNs in the shortest conductive pathway. After proper mathematical approximation, the relationship between the apparent ohmic resistance $R$ and strain $\varepsilon$ can be expressed as:

$$\ln R = -\frac{B(N_0-1+D)L_0}{U\lambda}\varepsilon + 2\ln\left(\varepsilon + \frac{d_0}{L_0}\right) + \ln U - F \quad (4)$$

$D$ and $F$ are both constants (detailed derivation of $D$ and $F$ are discussed in Supplementary Notes 2). As shown in Fig. 3e, Eq. (4) fits well with the experimental data. Moreover, the equation also fits well with the corresponding sensing curves when we fabricated the sensor with SCNs of different diameters (200 nm, 1 μm) (Supplementary Fig. 9), demonstrating the validity of the proposed model. For the dimensions of the present SCNs, $\frac{d_0}{L_0} > 0.138$ in this model (Supplementary Notes 3). Thus, $\ln(\varepsilon + \frac{d_0}{L_0})$ displays an approximate linearity with the applied strain ($R^2 > 0.95$) when the applied strain is between 0

and 60%. Moreover, $-\frac{B(N_0-1+D)L_0}{U\lambda}\varepsilon$ is a linear term, which means that the $\log R$-$\varepsilon$ curve is expected to show a linearity with $R^2 > 0.95$, theoretically. This model explains the observed high logarithmic linearity in our sensor, which reaches 0.992. In addition, according to this model, the sensitivity of the sensor is positively related to $-\frac{1}{\lambda}$, that is, the sharpness of the spikes. Thus, the $\lambda$ value of SCNs is larger than that of CB or randomly oriented CNTs due to the surface spiky structure, explaining why the SCNs-based sensor exhibits a larger sensitivity than its counterparts, as shown in Fig. 3d.

The third feature of present sensor is the good repeatability. After the first several stretch-release cycles whose resistance change is slightly higher than the following cycles (Supplementary Fig. 10)[57,60], the electrical signals become stable even after 5000 loading-unloading cycles at 10% strain (Fig. 3f). Additionally, even at 60% strain, the electrical signals can still maintain its stability and logarithmic linearity ($R^2 = 0.997$) after cyclic loading (Supplementary Fig. 11). The good repeatability observed is attributed to the sensing mechanism. In percolation- and contact resistance-based mechanisms, the resistance changes are often resulted from the destruction of conductive pathways. After multiple cycles, the conductive pathway may get permanently altered and cannot recover[61,62]. In contrast, in our case, the resistance change solely depends on the inter-distance increase of the SCNs rather than the destruction of conductive pathway, leading to an improved repeatability. Due to the viscoelasticity of the PDMS matrix and the friction between PDMS and the SCNs, the sensor exhibits electric signal hysteresis (Supplementary Fig. 12), which increases with the applied strain.

## High-density sensor arrays

The SCNs were further assembled at water-air interface to ensure consistent response in all directions and all units. For SCNs to float on water, their combined buoyancy and surface tension forces need to exceed gravity (Supplementary Notes 4). Thus, a lower density and higher surface tension coefficient were desired. The hollow structure of SCNs contributed to the reduced density, while surface tension coefficient could be enhanced through graphitization. As shown in Supplementary Fig. 13, increasing the carbonization temperature to 1200 °C allowed all the spheres to float on water and form an organized and continuous film. In addition, the floating assembly process ensured a uniform thickness across the film. As shown in Fig. 4a, b and Supplementary Fig. 14, films prepared with six different concentrations exhibit a consistent thickness of $1.6 \pm 0.2\,\mu m$ and a square resistance of $75 \pm 6\,k\Omega$ per sq. This is because the maximum load-carrying capacity of the films solely depends on the density of SCNs and surface tension coefficient (Supplementary Notes 4). With the same density and composition, the number of floating SCN layers remains a constant. Additional SCNs sprayed on water only increased the film area rather than the thickness, maintaining consistency in the SCNs film.

The patterning of the SCNs film into high density array was achieved by laser-scribing. While other patterning method such as photolithography is also viable, laser scribing is selected here due to its simplicity. It allows fast removal of SCNs with high resolution in one step. For example, the school badge pattern of Fudan University with a diameter of 3.8, 1.0 and 0.5 cm were produced, demonstrating a large-scale patterning capability (Supplementary Fig. 15). Moreover, a $10\,cm \times 10\,cm$ array pattern with a unit size of $100\,\mu m$ can be easily obtained within 10 min (Fig. 4c), confirming capabilities of high-resolution patterning. To collect signals from the high density SCN array, the interconnects adopt a wiring design similar to that of thin-film transistor. Specifically, silver microflakes/PDMS composite based conductive inks were printed on the patterned sensor array in both longitude and latitude directions as interconnects[63,64] and pure PDMS was dispensed at the junction of longitude line and latitude line as insulators (Fig. 4d).

The above process allowed fabrication of sensor arrays with different unit numbers and sensing densities (Supplementary Fig. 16). Limited by the line width of printed interconnects, the largest sensing density of the present sensor array was 100 per $cm^2$, which is still an order of magnitude higher than the previously reported strain sensing arrays (Fig. 4e). With the development of printing technique for interconnect materials, the decrease in the line width of the interconnection will enable an even higher array density. As shown in Fig. 4h, our sensing film displays both superior high sensing density and sensitivity compared with previously reported sensor arrays, offering high measurement precision and high spatial resolution simultaneously. This capability allows for the detection of 1% or even 0.1% strain (Supplementary Fig. 17) as well as the characterization of large strain gradients, which is crucial for many applications such as strain concentration, damage localization or crack propagation. Besides, our sensing film exhibits a combination of high spatial resolution and high stretchability (Fig. 4i and Supplementary Table 1). This feature opens up the possibilities for identifying and monitoring large and complex deformation, particularly in biomechanics- related strain monitoring.

Moreover, the well-organized structure and consistent thickness of the SCNs layer in the array ensure nearly identical sensing performance across all the sensing units of the array. To demonstrate this, electrical signals were gathered from 36 randomly chosen sensing units within a $30 \times 30$ array (100 pixel $cm^{-2}$) after applying a uniform 10% stretch to the device. As shown in Fig. 4f, the average recorded strain is 10.14%, which is closed to the applied strain. Notably, the SD among the 36 units is only 3.82%, indicating a high precision in strain field mapping. In order to further verify the stability of the process, 10 sensor arrays ($30 \times 30$, 100 pixel $cm^{-2}$) from the same batch of SCNs were prepared and the experiment in Fig. 4f was replicated. The average measured strain ranges from 10.02% to 10.26%, with the SD remaining within 3.82% across the ten array films (Fig. 4g and Supplementary Table 3). This yields a low standard error (SE) of 0.169%, thus showcasing a process consistency (detailed raw data can be found in Supplementary Fig. 18).

## Strain field mapping

With high sensitivity, high spatial resolution and good unit consistency, the strain sensing film can measure and reconstruct the strain fields. We tested its performance first in basic deformation modes and then expanded to more complex scenarios.

First, the sensing film was attached to a cuboid that was subjected to a uniaxial tension force. The accuracy of the reconstruction results was verified using FEA simulation. As shown in Fig. 5a, the sensing film shows a uniform strain field consistent with the simulation result. We further applied the sensing film to an object with gradient thickness, also subjected to uniaxial strain. The measured strain field shows a gradient strain distribution, similar to the FEA simulation (Fig. 5b).

To meet the demand for practical applications, we also tested the sensing capability in more complex situations. Particularly, we focused on crack propagation, which is a common failure mode in engineering applications. Although the previously reported sensor arrays were able to track the crack propagation process, the detailed picture of the large strain gradient distribution near the crack tip had never been captured before, mainly due to the low spatial resolution. Here the strain distribution near a crack tip was measured using our sensing film as a demonstration. A crack was created at the edge of an elastomer substrate and the sensing film was attached 1.5 mm away from the crack to monitor the strain distribution when a tensile strain was applied at the perpendicular direction (Fig. 5c). The measured strain distribution clearly reveals the concentrated strain at the crack tip and decreased from the tip to the outside, forming a strain gradient. Also, the strain distribution was symmetrical along the crack direction, exhibiting a heart shape. The above results match the simulation

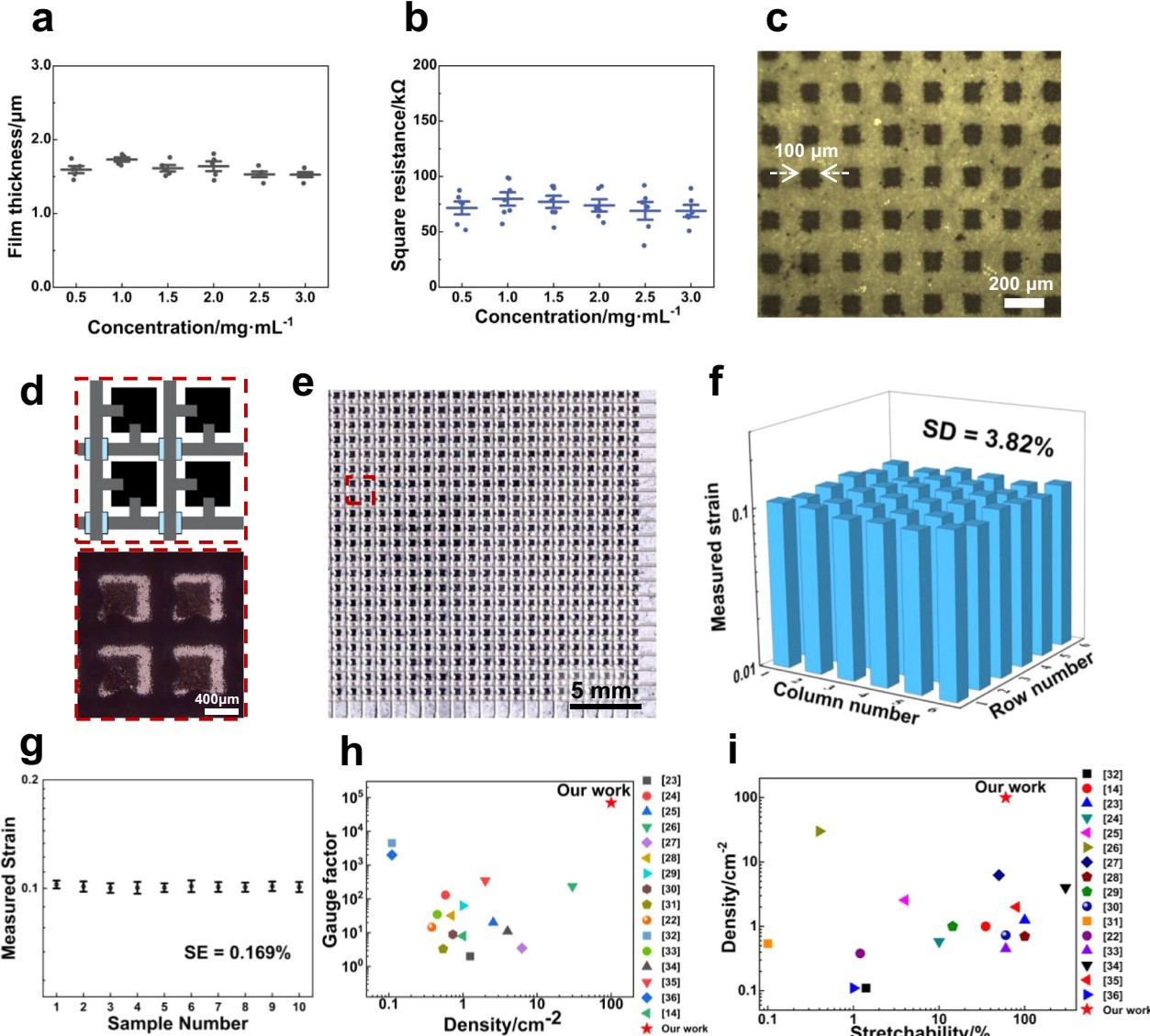

**Fig. 4 | Characterizations of high-density sensor array film. a** The average film thickness and (**b**) the average square resistances of SCN/PDMS composite films obtained with different SCN concentrations. The error bars represent standard deviation. **c** The SCNs films were patterned SCNs arrays with unit size of 100 μm. **d** Schematic diagram (up) and optical microscopic image (down) of four sensing units after interconnection. In the schematic diagram, the black squares are sensor units, the gray lines are interconnection lines made of silver/PDMS composites, and the blue squares are pure PDMS to separate longitude lines and latitude lines. **e** Photograph of sensor array with a density of 100 pixel cm⁻². **f** The response of 36 random sensing units when the sensing film is under a uniform strain of 10%. SD represents standard deviation. **g** The average measured strain from 10 sensor array films (30 × 30, 100 pixel cm⁻²) under a uniform strain of 10%. For each film, electrical signals from 36 units were collected. The error bars represent the standard deviation from each sensor array film and SE (standard error) is calculated from the results of the 10 sensor array films. **h** The sensor density and gauge factor of our sensor array as compared to the previously reported ones[14,22–36]. **i** The sensor density and stretchability of our sensor array as compared to the previously reported ones[14,22–36]. Source data are provided as a Source Data file.

results well, indicating that the sensing film can provide a simple and direct way to describe the detailed strain distribution of crack in structure health monitoring.

Indentation is another common anomaly in engineering applications. Drawing inspiration from Hans Christian Andersen's famous fairy tale The Princess and the Pea, where a princess had very sensitive skin that could feel a pea under twenty layers of mattress and layers of duck-down beds. Because of the sensitivity and high sensor density, here we show that our sensing film can detect the strain distribution like this princess in the fairy tale. A mung bean was buried under a 40 mm × 40 mm × 0.5 mm PDMS mattress and our sensing film was placed on top to measure the strain distribution. The section above the bean experiences concentrated strain than other areas. As shown in Fig. 6a, when there is only a piece of PDMS mattress on top, a signal

difference is observed, implying large deformations on surface of the PDMS. When the thickness of the PDMS mattress is adjusted to 5 mm, 10 mm, 15 mm and even 20 mm, although the $\Delta R/R$ signal decreases with the mattress thickness because the surface deformation becomes smaller, the existence of the bean can still be clearly distinguished. The corresponding strain field distributions were also recorded (Supplementary Fig. 19). With the high GF and high sensor density, our strain sensing film can be as sensitive as the princess in fairy tale.

Further, we demonstrated that our sensing film can measure the strain field not only in a static situation, but also in dynamic situations. Practically, a real-time data acquisition system was established to facilitate the real-time strain measurement (Supplementary Fig. 20a). The sensing film was conformally attached to the surface of a hardwood board, a soft sponge, and a pork slice, respectively. When we

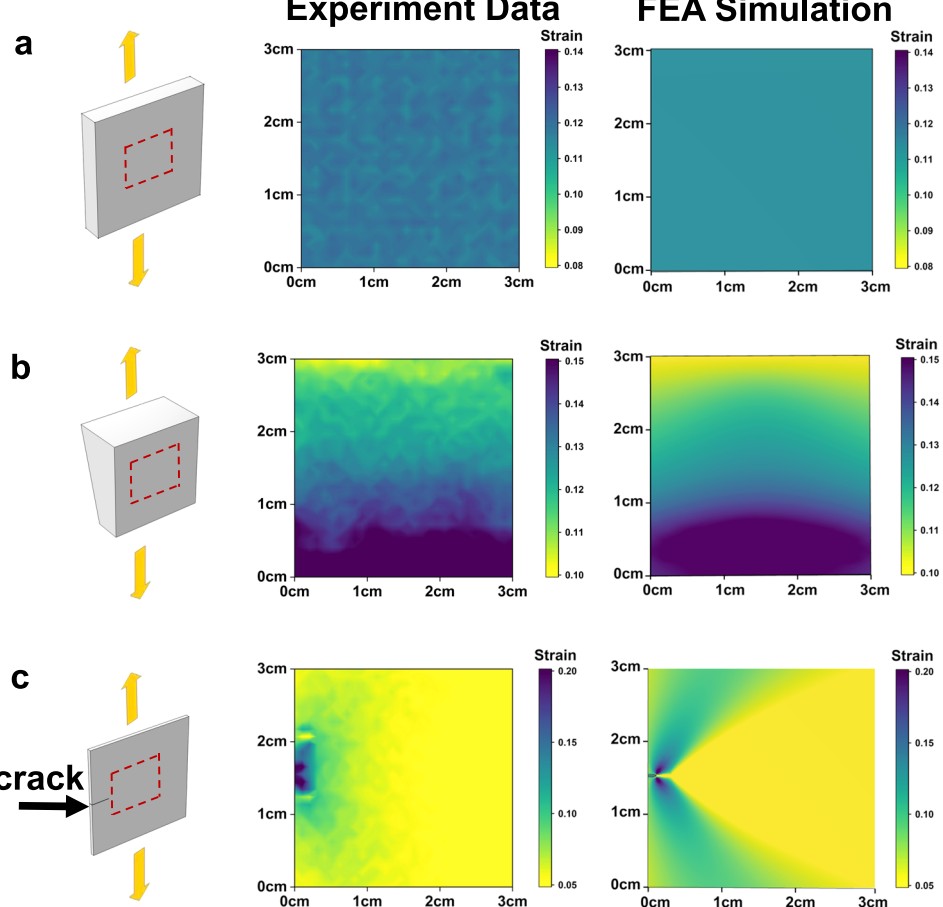

**Fig. 5 | Strain field reconstruction with our sensing film in different deformation states. a** The surface is subjected to homogeneous uniaxial stretching (FEA: finite element analysis). **b** The tested object has a gradient thickness and is subjected to uniaxial stretching. **c** The sensing film is used to detect the strain distribution near a crack tip. All these test data were collected by a sensing array with a scale of 30 × 30 and a density of 100 pixel cm⁻². Source data are provided as a Source Data file.

poke the sensors attached to the hardwood board, the sensing film does not respond (Fig. 6b, Supplementary Movie 3). However, when we poke the one attached to the soft sponge, a signal change can be observed (Fig. 6c, Supplementary Movie 4), indicating that the signal change of our sensors is solely from the applied strain rather than from the compressive stress. The pork slice was used to simulate the measurement of deformation in biological tissues (Supplementary Movie 5). As illustrated in Fig. 6d and Supplementary Movie 6, when the pork slice is deformed, the signals from sensing film can accurately reflect the deformation process in real-time.

## Discussion

Strain field mapping, which offers more comprehensive information compared to discrete strain gauges, requires a sensor array with high sensor density, sensitivity, and consistency among sensor units. However, achieving all these characteristics simultaneously is very challenging. In this study, we have developed a PDMS-based sensing film using self-assembled and patterned spiky carbon nanospheres as the sensing units. Leveraging the F-N tunneling effect of the ordered spiky carbon nanosphere array, our strain sensor unit exhibited high sensitivity (GF = 70,000), high logarithmic linearity ($R^2$ = 0.992) over a large strain range of 0–60% and good repeatability. More importantly, the floating assembly process allowed us to achieve a sensor density one order of magnitude higher than previous reports with good unit consistency (SD ≤ 3.82%). With these properties, our sensor array can detect strain gradients in materials, track minor and large deformations, and monitor strain distributions in real-time. The sensing film offers a convenient approach for measuring strain fields in various applications such as flexible electronics, soft robotics, biomechanics, and structure health monitoring of civil and aerospace infrastructures.

## Methods

### Materials

Styrene (analytical grade) and absolute ethanol (analytical grade) were purchased from Shanghai Dahe Chemical Co., Ltd., and the styrene monomers were purified in a neutral alumina column before use. Azobisisobutyronitrile (AIBN, 99%), polyvinyl alcohol (PVA, 98%, Mw = 130,000) were obtained from Shanghai Aladdin Biochemical Technology Co., Ltd. Aniline (ANI, 99.5%, analytical grade), $Fe(NO_3)_3 \cdot 9H_2O$ (98.5%, analytical grade) were obtained from Sinopharm Chemical Reagent Co., Ltd. PDMS (Sylgard™ 184) was bought from Dow Chemical Company. Carbon black (CB,Vulcan XC72) and carbon nanotubes (CNTs) were provided by Shanghai Kai Yin Chemical Co., Ltd. and Suzhou Hengqiu Technology Co., Ltd., respectively. Silver flakes were bought from Sichuan Zhongchai Great Wall Precious Metals Co., Ltd. (SF-01C). KI was bought from Henan Daheweiye Chemical Co. All raw materials were used without further purification.

### Synthesis of SCNs

A 250 mL of three-necked flask was evacuated and replaced with nitrogen in advance to remove oxygen. Under the protection of nitrogen, 20 g of styrene monomer, 0.2 g of azobisisobutyronitrile, 1.8 g of PVA, 60.4 g of ethanol and 7.6 g of de-ionized (DI) water were successively added to the flask, the solution was stirred at 900 rpm for

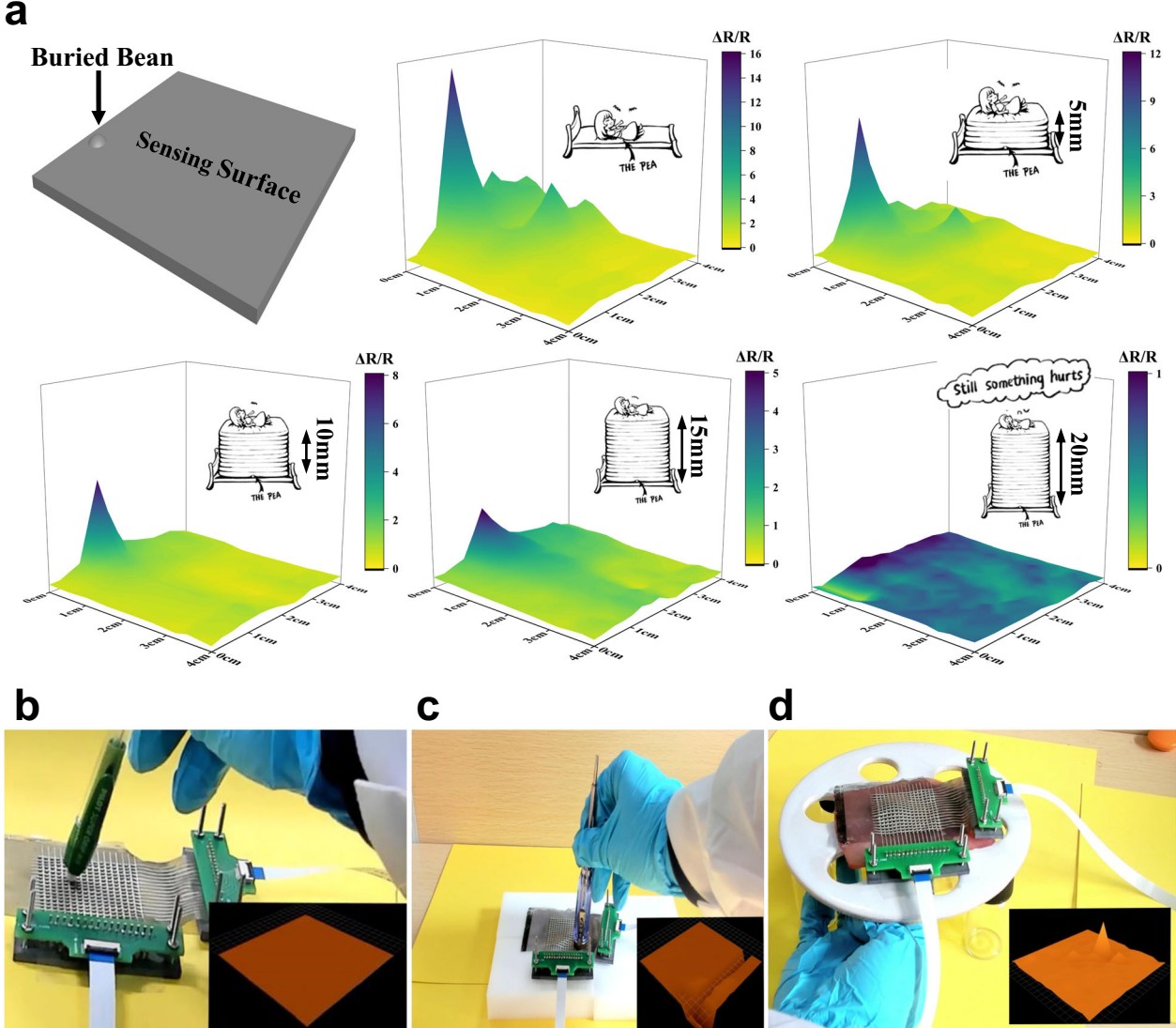

**Fig. 6 | Strain field monitoring and real-time measurement with our sensor film.** **a** The sensing film could detect even the tiny deformation induced by an embedded bean even when PDMS buffering layers of different thickness were on top. The film was as sensitive as the princess in Hans Christian Andersen's famous fairy tale The Princess and the Pea. **b** The sensing film did not respond to the poking when attached to a hardwood board. **c** The sensing film responded to the poking when attached to a soft sponge. **d** The deformation of a pork slice could be detected in a real-time manner when the sensing film was attached on it. To match with the dynamic data acquisition setup, the measurements were conducted using a 16 × 16 array with a density of 16 pixel cm$^{-2}$. Source data are provided as a Source Data file.

1 h at room temperature before refluxing at 90 °C for 12 h to obtain the PS nanosphere emulsion. These nanospheres were washed with ethanol and DI water (V/V = 1:1) by centrifuge and re-dispersed in DI water. 20 mL PS emulsion with a solid content of 0.3 g and 0.6519 g ANI monomers were mixed and stirred at 150 rpm for 5 h at room temperature. Then, 84 mL 0.5 M of $Fe(NO_3)_3$ aqueous solution was added to initiate the oxidative polymerization. The reaction was conducted at room temperature for 24 h to obtain a blue-black PS@PANI nanosphere solution, which was then centrifuged, washed with DI water and ethanol (V/V = 1:1), and finally freeze dried. The obtained PS@spiked PANI nanospheres were carbonized in a tube furnace under the protection of nitrogen, the temperature was increased from room temperature to 800 °C at 5 °C min$^{-1}$, then kept at 800 °C for 60 min, and further increased to 1200 °C at 2 °C min$^{-1}$. Finally, the temperature was kept at 1200 °C for 240 min to obtain the SCNs.

### Preparation of SCNs films

SCNs were dispersed in ethanol, and the dispersion was injected on the water surface drop by drop. After injection reached a certain volume, a loose layer of SCNs was formed at the gas-liquid interface. Then, a porous sponge was inserted into one side of the interface to quickly siphon water from the system, which drove the SCNs layer to pack along the opposite direction of the siphon. Closely packed SCNs films were obtained at the gas-liquid interface.

### Fabrication of the strain sensor array films

A sacrificial PVA substrate was prepared by scraping PVA aqueous solutions (13 wt%, Mw = 130,000) on a glass slide in advance. After drying, the slide was put into the container with closely packed SCNs film on water surface. The SCNs film was automatically transferred to the PVA substrate, followed by a $N_2$ drying process. Then, sensing array patterns with sensor units of 4 × 4, 8 × 8, 16 × 16, 25 × 25, and 30 × 30 were engraved on the SCNs films by laser scribing (EP-25-TGH-S, Han's Laser Technology Industry Group Co., Ltd., China). The laser marking and skip speed were 500 mm s$^{-1}$ and 1000 mm s$^{-1}$, respectively. Q frequency was 10 KHz, Q release was 5 μs, and the current was 20 A. Supplementary Movie 7 shows that the process of patterning a 30 × 30 array with a unit size of 500 μm needs to take about 80 s. After

patterning, a layer of PDMS (Sylgard™ 184, A:B = 10:1) was spin-coated on the sample with a spinning rate of 500 rpm for 20 s and 1500 rpm for 30 s, and then cured in an oven at 100 °C for 1 h. The spin coated PDMS layer maintains a thickness of 60 μm (Supplementary Fig. 4). The PVA-SCNs-PDMS sandwiched structured film was further placed in boiling water for 5 h to remove PVA sacrificial layer. Finally, Ag/PDMS conductive composite was printed as the conductive interconnect and pure PDMS was dispensed as the insulators. 2 g silver flakes were added to a beaker with 50 mL ethanol and 1 mL DI water, and stirred for 20 min. A certain amount of potassium iodide (KI) aqueous solution (0.01 mol/L) was added to the beaker and reacted for 15 min to remove the surfactant. After the reaction, the silver flakes were cleaned with ethanol before dried in vacuum. The iodized silver flakes were then exposed to light illumination (AM1.5 G illumination at 100 mW/cm$^2$ irradiation to simulate 1 sun) with a Newport−Oriel (Sol3A Class AAA Solar Simulator, 94043 A) solar simulator for a certain period of time and then cleaned with DI water and ethanol, respectively. The treated silver flakes could be dried and stored for the preparation of silver composites. Then, silver flakes were mixed with PDMS (A:B = 10:1) at a weight ratio of 3:1 to prepare conductive composite[63,64]. After curing the Ag/PDMS and pure PDMS, the sensor array film was obtained.

**Data acquisition and visualization system**

Supplementary Fig. 20b illustrates the principal schematic of the real-time data acquisition system for the sensor array. The units of array are arranged in a row × column configuration. Individual rows were sequentially enabled by driving the selected row line to VDD using a p-channel metal oxide semiconductor field effect transistor (MOSFET). Simultaneously, currents on all column lines were amplified using trans-impedance amplifiers composed of operational amplifiers (OPA4314, Texas Instruments) and feedback resistors (5.1 MΩ). The amplifiers' outputs were then sampled and digitalized by analog-to-digital converters (ADC, AD7091R, Analog Devices). Each ADC served eight column lines with a multiplexer. Overall system control was carried out by a microcontroller (MCU, STM32F407ZGT6, STMicroelectronics). The MCU transmitted data to a personal computer, The MCU transmitted data to a personal computer, where a custom code based on Python 3.9 pyqtgraph module facilitated real-time visualization. An adapter (Supplementary Fig. 20c) is used to facilitate the electrical interconnection between the sensing film and the data acquisition system. This adapter comprises two components: the first consists of a row of pogo pins, customized to align with the density of the sensing film array. The second component serves to transmit the signals from pogo pins to a cable, thereby establishing the interconnection with the data acquisition circuit.

**Characterization**

Morphologies of PS@PANI nanospheres, and SCNs were characterized by cryo-field emission scanning electron microscopy (FE-SEM, Zeiss Gemini SEM500, Germany) and transmission electron microscopy (TEM, HT7800, Hitachi, Japan). Images of the CB, CNT, CB films, CNT films, SCNs/PDMS composite films were obtained by a Phenom Prox desktop SEM (Phenom Prox, Netherlands). For SCNs/PDMS composite films, the samples were cut and prepared after freezing in liquid nitrogen. Optical photographs of the samples were acquired by an optical microscope (LEICA DFC425). The degree of graphitization of SCNs was characterized by XPS (PHI5300, PHI) and Raman spectroscopy (532-nm laser source, XploRA, HORIBA JobinYvon). The resistance was measured by a Keithley 6517B and mechanical properties were obtained using an electronic universal testing machine (CMT-4304-QY, Zhuhai Sansitaijie Electrical Equipment Co., Ltd., China). Square resistances of the composite film samples were obtained by four probe equipment (SB120, Shanghai Qianfeng Electronic Instrument Co., Ltd., China). The commercial FEA software package,

COMSOL Multiphysics (Sweden), was used for FEA simulation. A hyperelastic model based on Neo-Hookean was used as the constitutive model. All models were performed according to the actual sample situations.

**Reporting summary**

Further information on research design is available in the Nature Portfolio Reporting Summary linked to this article.

## Data availability

Source Data file has been deposited in Figshare database under accession code https://doi.org/10.6084/m9.figshare.25020161[65]. Source data are provided with this paper.

## Code availability

The custom code used to analyze and plot the measurement data from circuit in real time has been deposited at GitHub. The Github repository was linked to Zenodo and can be found at https://doi.org/10.5281/zenodo.10517188[66].

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

## Acknowledgements

We appreciate the financial support from the National Key Research and Development Program of China (2022YFA1205200, L.W.), National Natural Science Foundation of China (U22A20249, 52173071, Z.L.), National Natural Science Foundation of China (52188102, J.G.) and Postdoctoral Science Foundation of China (2021M690648, S.M.). We also thank Ms. Yang Shuo for her guidance in photography.

## Author contributions

L.W., Z.L., S.M. and H.Y. conceived the concept and designed the research. S.M., H.Y. and J.Z. conducted the experiments. Y.Q. and Y.J. conducted the 30 × 30 detection array circuit design. S.M., H.Y., wrote the manuscript. L.W. and Z.L. revised the manuscript. Y.X., L.S. and J.G. discussed the results and commented on the manuscript.

## Competing interests
The authors declare no competing interests.
