## [Peer Review File · Nature Communications]

REVIEWER COMMENTS

Reviewer #1 (Remarks to the Author):

This manuscript presents the fabrication of a highly sensitive strain sensor by using the tunneling effects at the interface of spiky carbon nanospheres. The outstanding sensitivity of the strain sensor, demonstrated by these structures, and the practical application via the realization of high-density sensor arrays, aligns with the esteemed standards of Nature Communications. Such advances will match the interest of its readership. However, it is imperative that certain non-scientific nuances, albeit minor, be rectified. Furthermore, there are issues that the authors must address, as described below.

Major comments

1. The dual sweep characteristics should be included, including forward and reverse sweeps of an applied strain. Many stretchable devices based on polymers or elastomer substrates have suffered from mechanical hysteresis.
2. Transient responses (showing the delay times for the stretching state and releasing state) are required.
3. Compared to the operational range of 60% and resistance change magnitude of 10 Gohm presented by the authors, the 10% operational range and resistance change of around 1 Mohm shown in Figure 3f seem limited. It might be beneficial to consider providing data from repeated measurements at large strain ranges, even if only for a significantly reduced number of cycles.
4. The authors should clarify if any disparity in sensing was observed between the device's initial form and its subsequent cycles. The citation on line 218 touches upon pre-stretching. However, its core premise is centered around the generation of low-resistance through the elasticity of pre-stretched elastomers, not the alleviation of internal elastomer stress via stretching.
5. The authors conducted the strain test up to 60% stretching. Beyond this range, is the electrically conductive path disconnected or remains unchanged?
6. The tensile strain to break off a PDMS elastomer is known to be ~100% strain, but it can easily tear even in the 60% range of tensile strain, similar to the test performed by the authors. Is the 60% detection range that the authors mentioned due to the SCN sensing material or the substrate? Can the detection range be extended by using other compliant materials such as Ecoflex as a substrate?
7. Figures 3b and 4h-i employ geometric delineations such as circles and rectangles. I wonder if these shapes denote the theoretical confines of the sensing mechanism as extrapolated via principal component analysis (PCA). If this is just an explanatory symbol, I recommend removing it to avoid confusing the reader. If these aren't merely illustrative devices, it might be prudent to elucidate the underlying theoretical impetus for their inclusion.

8. Arraying via laser scribing seems to be a time-consuming and serial process. Please specify the time taken to fabricate a single array and add a discussion regarding improving scalability such as patterning methods exploiting photolithography technologies.

9. The author should state a numerical value for the "mild strain" in lines 274-276 which seems to be not a scientific statement and add a relevant reference.

Minor comments

1. 1. Phrases such as "which have never been reported before" in lines 71, 148-149, and 277-278 are excessive claims.

2. Is the terminology in Line 100 "calcination" correct? It was not used in the cited reference and the authors' previous work.

3. Add a reference when introducing the tunneling equation in line 184.

Editorial errors

1. Check spacing including "1200 degree Celsius" in line 100, Figure 2, lines 166-167, and line 208.

2. Full names for abbreviations like "PANI", "SEM", and "Ag" have not been provided.

3. Use more scientific terminologies, than "hot water" and "ohmic type".

4. Omit the unit "mm" in Line 346.

5. Figures S13 and S14 both appear twice. Please double-check the figure numbering.

Reviewer #2 (Remarks to the Author):

- Please comment on the response time of the sensor.
- What is the percolation threshold of spiky carbon nanospheres?
- In the fabrication of sensor array, what is the spin speed and thickness of PDMS on the patterned film?
- Some studies suggest that the resolution of the entire sensor array can be improved by using conductive threads/wires as electrodes. How can the current study include conductive threads as electrodes?
- Can the present sensor array be used for the wider detection range? Have the authors tested the sensor beyond 60% strain?

Reviewer #3 (Remarks to the Author):

The current manuscript entitled “High-density, highly sensitive sensor array of spiky carbon nanospheres for strain field mapping” demonstrates a well-thought-out experimental design, reasonable result characterization, and an innovative sensor configuration. The abundant supporting information, in particular, is highly persuasive. I wholeheartedly support the publication of this research in Nature Communications. However, before publication, there are some issues that both the authors and the editor should consider for potential revisions:

1. The current conclusions may be in error. In line 388, it is mentioned that the sensor demonstrates linearity within the 0-60% strain range. However, based on Figure 3a, it is evident that the sensor exhibits exponential behavior, as this figure employs logarithmic coordinates. This discrepancy raises concerns regarding the stated linearity, and it is imperative that this issue is addressed and clarified before publication.

2. In lines 169 to 171, the authors mention that with an increase in voltage, resistance decreases; however, they do not provide a detailed explanation. The current explanation can only account for why the rate of resistance reduction is greater at 60% strain compared to 0% strain. Regarding the phenomenon of voltage-induced resistance reduction, I recommend that the authors provide a more comprehensive explanation. Additionally, I suggest that the authors consider observing the time-dependent changes in the reported strain sensor resistance at 0% strain under a fixed 40V voltage, as this may help elucidate the underlying mechanisms.

3. In Figure 3d, it is indicated that the red line representing SCNs (though noted as SCS in the figure) exhibits a more pronounced increase in resistance with increasing strain compared to the CB-based sensor. This phenomenon requires an explanation, considering that the two have similar dimensions and structures, and both can be described by similar physical models when considering tunneling effects – as both are spherical 0-dimensional entities.

Point-by-Point Responses to Reviewers' Comments

To Reviewer # 1

General comments: This manuscript presents the fabrication of a highly sensitive strain sensor by using the tunneling effects at the interface of spiky carbon nanospheres. The outstanding sensitivity of the strain sensor, demonstrated by these structures, and the practical application via the realization of high-density sensor arrays, aligns with the esteemed standards of Nature Communications. Such advances will match the interest of its readership. However, it is imperative that certain non-scientific nuances, albeit minor, be rectified. Furthermore, there are issues that the authors must address, as described below.

General answers: We thank the Reviewer very much for his or her positive comments and valuable suggestions that are immensely helpful in enhancing the quality of our manuscript. The issues you are concerned are addressed as follows:

Major comments

Q1: The dual sweep characteristics should be included, including forward and reverse sweeps of an applied strain. Many stretchable devices based on polymers or elastomer substrates have suffered from mechanical hysteresis.

A1: The hysteresis tests at different strains have been added. Experimental results indicate that our sensor does exhibit hysteresis during usage and it becomes more pronounced when the applied strain increases (Figure A1). As the Reviewer mentioned, this hysteresis phenomenon is not uncommon in other elastomer-based strain sensors, some of the examples are shown in the following links. This may be caused by the viscoelasticity of the PDMS matrix and the friction between PDMS and the SCNs.

ACS Nano: <https://doi.org/10.1021/acsnano.1c00259>

ACS Nano: <https://doi.org/10.1021/acsnano.8b07805>

While completely eliminating the hysteresis in our sensor might pose a challenge, we can mitigate its influence on strain measurements through signal processing. In particular, by calculating the derivative of the resistance with respect to time, we can address this issue. A positive derivative suggests the sensor is in the stretching stage, allowing us to determine the strain value based on the sensing curve during stretching.

Conversely, a negative derivative signals the releasing stage, and the strain value is determined from the response curve during unloading. The above discussion is added in the blue words in Page 10 and Supplementary Fig. 12 in the revised manuscript.

Figure A1. Hysteresis tests of SCNs/PDMS sensor (a) 20% strain; (b) 30% strain; (c) 40% strain; (d) 50% strain; (e) 60% strain.

Q2: Transient responses (showing the delay times for the stretching state and releasing state) are required.

A2: We stretched our sensor for about 2 mm (about 15 % strain) and maintained the strain for about 10 s before release. The response time is 80 ms and 110 ms for the stretching state and releasing state, respectively (Figure A2). Please see the blue words in Page 7 in the revised manuscript and Supplementary Fig. 6.

Figure A2. Response time of SCNs/PDMS sensor.

Q3: Compared to the operational range of 60% and resistance change magnitude of 10 Gohm presented by the authors, the 10% operational range and resistance change of around 1 Mohm shown in Figure 3f seem limited. It might be beneficial to consider providing data from repeated measurements at large strain ranges, even if only for a significantly reduced number of cycles.

A3: Thanks for the advice. We agree that repeated measurement at large strain is meaningful to improve our work. Thus, we have extended the strain range of the cycle test from 10% to 60% (Figure A3). Through experiments, we observed that even under conditions of 60% strain, our sensor maintains good repeatability, and the logarithmic linearity ($R^2=0.997$). Please see the blue words in Page 9 in the revised manuscript and Supplementary Fig. 11.

Figure A3. 60% cyclic test on SCNs/PDMS sensor.

Q4: The authors should clarify if any disparity in sensing was observed between the device's initial form and its subsequent cycles. The citation on line 218 touches upon pre-stretching. However, its core premise is centered around the generation of low-resistance through the elasticity of pre-stretched elastomers, not the alleviation of internal elastomer stress via stretching.

A4: Thank you for the careful examination of our work. We sincerely apologize for the misunderstanding caused by inappropriate reference on line 218. When we mentioned "pre-stretching," we referred to the first five to ten stretch-release cycles whose resistance change is slightly higher than the following cycles. After these initial cycles, the sensor entered a stable state and the resistance change is highly repeatable. Figure

3f in the manuscript exhibited the data of all 5000 cycles and the difference of the first several cycles is not obvious. When we zoom in the first 50 cycles as shown in Figure A4, the difference is clear. This phenomenon is well-documented, as shown in some recent reports on elastomer-based flexible strain sensors (ACS Appl. Mater. Interfaces. 2018, 10, 4, 3948–3954; Adv. Mater., 2016, 28(31): 6640-6648).

To avoid the confusion, we changed the wording in the revised manuscript from “pre-stretching” to the “first several stretch-release cycles” in Page 9 in the revised manuscript and added Figure A4 to Supplementary information as Supplementary Fig. 10. Also, we replaced the original reference with the above two papers in the revised manuscript.

Figure A4. The first 50 cycles from the 5000 cycles test in manuscript.

Q5: The authors conducted the strain test up to 60% stretching. Beyond this range, is the electrically conductive path disconnected or remains unchanged?

A5: Once the strain exceeds this range, the resistance is so high that it reaches the upper limit of the high resistance meter (6517B, Keithley, American). Although the claimed upper limit is 200 T Ω , this can only be achieved with the assistance of the

electromagnetic shielding fixture 8001 (Reference manual is available at: <https://www.tek.com/en/manual/specialty-instruments/keithley-high-resistance-low-current-electr-0-electrometers>). In our case, as we needed to stretch the sensor, we couldn't secure the sensor in the 8001 fixture. Consequently, the testing upper limit of the high resistance meters reduced to 100 G Ω , as evidenced by the open-circuit resistance reading of 100 G Ω . Similarly, our sensor's signal also ceased to change after approaching 100 G Ω (Figure A5). This is also the reason we did not include data above 60% strain. Please see the added explanation in blue words in Page 6 in the revised manuscript and Supplementary Fig. 5.

Figure A5. (a) 100% strain cyclic test of SCNs/PDMS sensor. The reading stops to change after reaching 60% strain but it can return to the original value after the tensile strain is reduced. (b) Open-circuit resistance of the high resistance meter.

Q6: The tensile strain to break off a PDMS elastomer is known to be ~100% strain, but it can easily tear even in the 60% range of tensile strain, similar to the test performed by the authors. Is the 60% detection range that the authors mentioned due to the SCNs sensing material or the substrate? Can the detection range be extended by using other compliant materials such as Ecoflex as a substrate?

A6: We thank the Reviewer for this question. As mentioned in Q5, our sensing range is limited by the measurement range of the high-ohmic meter rather than the stretching range of the substrate. As shown in Figure A6, although replacing PDMS substrate with ecoflex is helpful to enhance the tearing strength, the final state after stretching is

similar, with signals reaching a plateau because of the measurement limit. Therefore, we only report the data up to 60% strain.

Figure A6. 100% strain cyclic test of SCNs/Ecoflex sensors.

Q7: Figures 3b and 4h-i employ geometric delineations such as circles and rectangles. I wonder if these shapes denote the theoretical confines of the sensing mechanism as extrapolated via principal component analysis (PCA). If this is just an explanatory symbol, I recommend removing it to avoid confusing the reader. If these aren't merely illustrative devices, it might be prudent to elucidate the underlying theoretical impetus for their inclusion.

A7: We thank the Reviewer for the good advice. These geometric delineations only serve as guides to the eye and do not have any scientific indications. Therefore, to avoid the confusion, we have removed those from the Figure 3b; 4 h and 4i as suggested by the Reviewer.

Q8: Arraying via laser scribing seems to be a time-consuming and serial process. Please specify the time taken to fabricate a single array and add a discussion regarding improving scalability such as patterning methods exploiting photolithography technologies.

A8: The time needed for laser scribing depends on the sample area. Here, samples are patterned by laser scribing (EP-25-TGH-S, Han's Laser Technology Industry Group Co.,

Ltd., China). The laser marking and skip speed were set at 500 mm/s and 1000 mm/s, respectively. Q frequency was 10 KHz, Q release was 5 μ s, and the current was 20 A. Patterning a 4 cm \times 4 cm array took approximately 80 seconds, as demonstrated in Supplementary Movie 7. Given that laser processing time is generally proportional to the processing area, the processing time increases as the array size expands. In contrast, the time required for patterning through photolithography does not depend on the size of the device. However, photolithography involves multiple steps, including photoresist coating, pre-exposure baking, exposure, post-exposure baking, development, etching, and photoresist removal. While laser-scribing is a one-step process, determining which route is more time-efficient remains a case-by-case consideration.

Additionally, applying photolithography to pattern the sensor array requires extensive work in process optimization. The high surface smoothness requirement for the photoresist layout poses a challenge, especially considering the increased surface roughness caused by the protruding SCNs on the substrate. This necessitates an augmented thickness of the photoresist layer, increasing both cost and processing difficulty. In the etching step, wet etching may not be feasible due to the potential capillary flow of the etchant into surrounding areas caused by cavities from the SCNs' spikes, thereby affecting processing accuracy. Dry etching, while viable, requires a parameter optimization study to ensure sufficient selectivity in etching the SCNs rather than the surrounding photoresist, considering both are primarily composed of carbon. Given these considerations, although photolithography offers higher resolution and potential time savings in large-scale fabrication, we chose laser scribing for its simplicity in the patterning process. A discussion on the fabrication time of the sensor array and the rationale behind selecting laser scribing for patterning has been added to Page 19 and Page 11 in the revised manuscript, respectively.

Q9: The author should state a numerical value for the "mild strain" in lines 274-276 which seems to be not a scientific statement and add a relevant reference.

A9: We appreciate the Reviewer's kind advice. To quantify the capability of our strain sensors for small strain detection, we record the resistance change of our sensor at 1%

strain and 0.1% strain, respectively. As shown in Figure A7, the sensor can detect the cyclic strain at 1% and 0.1%, and the resistance values are consistent with our prediction model listed in Supplementary notes 2. Thus, we believe that our sensor can measure the “mild strain” as low as 0.1%. It is to be noticed that our sensor may detect even lower strains. But restricted by the small sensor size and the accuracy of the universal testing machine (CMT-4304-QY, Zhuhai Sansitaijie Electrical Equipment Co., Ltd., China) used to apply the tensile stress, we only tested 0.1% strain. Even so, this result is comparable or even better than other reported strain sensors. Please see the added “mild strain” in page 12 and Supplementary Fig. 17.

Figure A7 The response of the SCNs/PDMS strain sensor to the cyclic tensile test at (a) 1% strain and (b) 0.1% strain.

Minor comments

Q1: Phrases such as “which have never been reported before” in lines 71, 148-149, and 277-278 are excessive claims.

A1: We have deleted those excessive claims.

Q2: Is the terminology in Line 100 “calcination” correct? It was not used in the cited reference and the authors’ previous work.

A2: Thanks for your advice. Calcination refers to the thermal treatment generally for the purpose of removing impurities or volatile substances. Here, our main focus in this process is to convert the PANI spiky nanospheres to carbon spiky nanospheres. Although this process is somewhat similar to calcination, the term “carbonization” may

be more suitable in this context. Therefore, we have changed the wording in the revised manuscript, and we appreciate the reviewer's suggestion.

Q3: Add a reference when introducing the tunneling equation in line 184.

A3: The corresponding reference is added.

Editorial errors

Q1: Check spacing including “1200 degree Celsius” in line 100, Figure 2, lines 166-167, and line 208.

A1: We have double checked the space issue and corrected it.

Q2: Full names for abbreviations like “PANI”, “SEM”, and “Ag” have not been provided.

A2: Full names have been provided for all these abbreviations when they appear for the first time.

Q3: Use more scientific terminologies, than "hot water" and “ohmic type”.

A3: The “hot water” has been replaced by “95°C water”, The “ohmic type” has been replaced by “ohmic conductor”.

Q4: Omit the unit “mm” in Line 346.

A4: The omitted unit “mm” is added.

Q5: Figures S13 and S14 both appear twice. Please double-check the figure numbering.

A5: We have double-checked the figure caption in supplementary data and corrected the figure caption in supplementary information.

To Reviewer # 2

Q1: Please comment on the response time of the sensor.

A1: We have tested the response time for our sensor. We stretched our sensor for about 2 mm (about 15 % strain) quickly and maintained the strain for about 10 s before release. The response time is 80 ms and 110 ms for the stretching state and releasing state, respectively (Figure A8). Please see the blue words Page 7 in the manuscript and Supplementary Fig. 6.

Figure A8. Response time of SCNs/PDMS sensor.

Q2: What is the percolation threshold of spiky carbon nanospheres?

A2: The operational principle of our sensor differs from that of percolation-type sensors. In percolation theory, conductive fillers are randomly dispersed in the matrix, and electrical conductivity experiences a gradual increase with filler loading until the formation of a conductive pathway leads to a dramatic rise in conductivity. In our approach, instead of random dispersion, SCNs undergo self-assembly, forming a closely packed thin film on the top surface of the PDMS substrate. This closely packed structure ensures that the electrical properties of the sensing layers remain almost unchanged with increasing quantities of SCNs (refer to Figure 4b).

Furthermore, our observations suggest that the tunneling effect arises from the SCNs exposed to air rather than those buried within the PDMS. Coating a PDMS layer on top of the sensor and burying all SCNs within the PDMS matrix results in a loss of conductivity. This can be attributed to the lower dielectric constant of air, facilitating tunneling more easily than within the PDMS matrix. In this context, the PDMS matrix primarily serves to provide stretchability. Therefore, the present system is not related to the percolation theory for composites and we cannot directly obtain the percolation threshold from the present study.

Q3: In the fabrication of sensor array, what is the spin speed and thickness of PDMS on the patterned film?

A3: PDMS (Sylgard™ 184, A:B=10:1) is used for spin coating without further dilution. The spin coating speed is 500 r/min for 20s and then 1500 r/min for 30s. The thickness of PDMS is around 60 μm as shown in Figure A9. We have added the detailed spin coating speed in page 19 in the revised manuscript and added the Figure A9 as Supplementary Fig. 4 in the Supplementary information.

Figure A9. The thickness of PDMS layer.

Q4: Some studies suggest that the resolution of the entire sensor array can be improved by using conductive threads/wires as electrodes. How can the current study include

conductive threads as electrodes?

A4: We appreciate the reviewer's suggestion. Indeed, the array density is constrained by the conductive line width and pitch, and enhancing the resolution of the entire sensor array is feasible with thinner conductive lines. For these lines, attributes such as metal-like conductivity, stretchability, small thickness, patternability, and strong bonding with the substrate are imperative. Currently, there are three materials that serve as the conductive lines for stretchable sensor arrays.

1) Thin metal foils with wavy structures (e.g. Nature Communications, 2022, 13:7757). Thin metal foils with wavy structures, such as serpentine structures, exhibit high conductivity and provide excellent stretchability. Patterning is achieved through photolithography, allowing for thin conductive lines ranging from several microns to several tens of microns. While offering the potential for higher sensor density, integrating these conductive lines with the SCNs arrays on the same soft substrate poses challenges. One option involves fabricating the conductive lines first and transferring them to the PDMS substrate with SCNs. However, bonding often involves plasma treatment and compression, and the protruding SCNs on the surface can impede the compression process. Another option is in-situ fabrication of conductive lines on the substrate, but the presence of SCNs may negatively impact the exposure and development process. Compared to previous sensor arrays based on thin and flat sensor films, the current system is less compatible with the conductive line fabrication process.

2) Conductive composites. In this study, we utilized conductive composites made of Ag microflakes/PDMS. These composites can be directly printed on the PDMS substrate without affecting the SCNs. Since both the substrate and conductive lines are PDMS-based, they form a stable bond even under stretching. Recent reports have used Ag nanowires/PDMS composites to form 20 micron wide lines with the assistance of photolithography (Science, 2021, 373:1022-1026.). However, due to concerns about the impact of SCNs on the photolithography process, this approach may not be compatible with our system.

3) Liquid metal lines (Nature, 2023, 613:667-675). Patterning liquid metal lines involves techniques like screen printing, and the resolution is comparable to the current

direct printing process used in the manuscript.

In sum, to achieve a high resolution, photolithography is usually required. But protrusion structure of SCNs on the substrate may affect the photolithography process. That is why we choose direct printing of Ag/PDMS composites.

Q5: Can the present sensor array be used for the wider detection range? Have the authors tested the sensor beyond 60% strain?

A5: We thank the Reviewer for this question. We did test the sensor beyond 60% strain, but once the strain exceeded this range, the resistance was so high that it reached the upper limit of the high resistance meter (6517B, Keithley, American). Although the claimed upper limit of the high resistance meter is 200 T Ω , this can only be achieved with the assistance of the electromagnetic shielding fixture 8001. In our case, as we needed to stretch the sensor, we couldn't secure the sensor in the 8001 fixture (Reference manual is available at: <https://www.tek.com/en/manual/specialty-instruments/keithley-high-resistance-low-current-electr-0-electrometers>).

Consequently, the testing upper limit of the high resistance meters reduced to 100 G Ω , as evidenced by the open-circuit resistance reading of 100 G Ω . Similarly, our sensor's signal also ceased to change after approaching 100 G Ω (Figure A10). This is why we did not include data above 60% strain. We have added the explanation in page 6 and Supplementary Fig. 5.

Figure A10. (a) 100% strain cyclic test of SCNs/PDMS sensor. (b) Opening resistance test using high resistance meter.

To Reviewer # 3

General comments: The current manuscript entitled “High-density, highly sensitive sensor array of spiky carbon nanospheres for strain field mapping” demonstrates a well-thought-out experimental design, reasonable result characterization, and an innovative sensor configuration. The abundant supporting information, in particular, is highly persuasive. I wholeheartedly support the publication of this research in Nature Communications. However, before publication, there are some issues that both the authors and the editor should consider for potential revisions:

General answers: We appreciate the Reviewer for his or her positive evaluation and valuable suggestions. We have carefully considered the reviewer's comments and made corresponding revisions in the manuscript as follows:

Q1: The current conclusions may be in error. In line 388, it is mentioned that the sensor demonstrates linearity within the 0-60% strain range. However, based on Figure 3a, it is evident that the sensor exhibits exponential behavior, as this figure employs logarithmic coordinates. This discrepancy raises concerns regarding the stated linearity, and it is imperative that this issue is addressed and clarified before publication.

A1: We apologize for the confusion of “linearity” in line 388. It is actually “logarithmic linearity” as we emphasized both in Abstract and Introduction sections in manuscript. Logarithmic linearity here means the sensor exhibits exponential behavior and can be transformed into linearity behavior by a single logarithmic operation on the signal. To avoid confusion of the expression, we changed the word “linearity” into “logarithmic linearity” in Page 18.

Q2: In lines 169 to 171, the authors mention that with an increase in voltage, resistance decreases; however, they do not provide a detailed explanation. The current explanation can only account for why the rate of resistance reduction is greater at 60% strain compared to 0% strain. Regarding the phenomenon of voltage-induced resistance reduction, I recommend that the authors provide a more comprehensive explanation. Additionally, I suggest that the authors consider observing the time-dependent changes

in the reported strain sensor resistance at 0% strain under a fixed 40V voltage, as this may help elucidate the underlying mechanisms.

A2: We thank the Reviewer very much for the good suggestion. The voltage-induced resistance reduction is a typical phenomenon for a tunneling-based sensor. The increased voltage increases the energy of electron and reduces the tunneling barrier, which is beneficial for tunneling, leading to a higher tunneling current (i.e., lower resistance). More specifically, the voltage-induced resistance reduction can be explained by the F-N tunneling model. In F-N tunneling model, the tunneling current I can be expressed as:

$$I = AE_d^2 e^{(-B/E_d)}$$

where E_d represents the electrical field strength. A and B are positive constant, Thus, with the applied voltage increasing, the current increases and the resistance decreases. Therefore, we used this test to confirm the tunneling effect in our sensor during the whole strain range (0-60%).

In addition, according to the Reviewer's suggestion, the time-dependent resistance test at 0% strain under a fixed 40V voltage was performed for an hour. As shown in Figure A11, the resistance remains stable with a variation less than 0.44% (standard deviation). Since F-N tunneling model does not contain a time-dependent term, the result here is in agreement with the model.

In sum, the resistance is only affected by the electrical field strength, which includes the applied electrical voltage and the inter-sphere distance (i.e. the strain) while not related to the time. Please see the blue words in Page 7 in the revised manuscript.

Figure A11. The resistance changes at 0% under 40 V

Q3: In Figure 3d, it is indicated that the red line representing SCNs (though noted as SCS in the figure) exhibits a more pronounced increase in resistance with increasing strain compared to the CB-based sensor. This phenomenon requires an explanation, considering that the two have similar dimensions and structures, and both can be described by similar physical models when considering tunneling effects – as both are spherical 0-dimensional entities.

A3: Thanks for the Reviewers' comment. We are sorry for the typo and have corrected the note from 'SCS' to 'SCN' in Figure 3d. Although SCNs and CB are both spherical 0-dimensional particles, their morphologies are quite different. The SCNs has a unique spiky structure (Figure A12a) while CB does not (Figure A12b). The spiky structure of SCNs is the key factor of our higher resistance increase. The spike on the surface of SCNs can enhance electrical field strength owing to its sharpness induced electrical field strength concentration effect (detailed mechanism can be checked in Supplementary Notes 1). Thus, the electrical field strength between SCNs particles is larger than that of CB particles under the same applied voltage. After taking this effect in account, the F-N tunneling based sensing behavior obeys the following equation (details can be checked in Supplementary Notes 2):

$$\ln R = -\frac{B(N_0 - 1 + D)L_0}{U\lambda} \varepsilon + 2 \ln \left(\varepsilon + \frac{d_0}{L_0} \right) + \ln U - F$$

Here, λ represents the electrical strength concentration coefficient. Its value is 1 for spherical particles without spike and become closer to zero with the sharpness of the spike increasing. So, SCNs based sensor can exhibit much higher gauge factor

compared to CB based sensor. Please see the more explanation in page 9 in the revised manuscript.

Figure A12. (a) SEM and TEM images of the SCNs. (b) SEM image of CB.

REVIEWERS' COMMENTS

Reviewer #1 (Remarks to the Author):

The authors addressed all the comments and issues I raised. Therefore, I believe that the paper is suitable for publication in nature communications.

Reviewer #2 (Remarks to the Author):

I recommend that this paper be accepted. Revisions are accepted.

Reviewer #3 (Remarks to the Author):

The authors provided highly detailed responses to my inquiries. From my perspective, this manuscript is entirely suitable for publication in your esteemed journal. I look forward to witnessing their further related research in the future.

Point-by-Point Responses to Reviewers' Comments

To Reviewer #1

General comments: The authors addressed all the comments and issues I raised. Therefore, I believe that the paper is suitable for publication in nature communications.

General answers: Thank you for your valuable feedback. We truly appreciate the time and effort you have put into reviewing our work.

To Reviewer #2

General comments: I recommend that this paper be accepted. Revisions are accepted.

General answers: Thank you for your valuable feedback. We truly appreciate the time and effort you have put into reviewing our work.

To Reviewer #3

General comments: The authors provided highly detailed responses to my inquiries. From my perspective, this manuscript is entirely suitable for publication in your esteemed journal. I look forward to witnessing their further related research in the future.

General answers: Thank you for your valuable feedback. We truly appreciate the time and effort you have put into reviewing our work.